# circRNA from *APP* Gene Changes in Alzheimer’s Disease Human Brain

**DOI:** 10.3390/ijms24054308

**Published:** 2023-02-21

**Authors:** Amaya Urdánoz-Casado, Javier Sánchez-Ruiz de Gordoa, Maitane Robles, Miren Roldan, Mónica Macías Conde, Blanca Acha, Idoia Blanco-Luquin, Maite Mendioroz

**Affiliations:** 1Neuroepigenetics Laboratory-Navarrabiomed, Complejo Hospitalario de Navarra-IdiSNA (Navarra Institute for Health Research), Universidad Pública de Navarra (UPNA), Pamplona, 31008 Navarra, Spain; 2Department of Neurology, Complejo Hospitalario de Navarra-IdiSNA (Navarra Institute for Health Research), Pamplona, 31008 Navarra, Spain

**Keywords:** Alzheimer’s disease, entorhinal cortex, circRNA, mRNA, Amyloid beta, *APP*

## Abstract

Alzheimer’s disease (AD) is the most common cause of age-related dementia. Amyloid precursor protein (*APP*) is the precursor of Aβ peptides, and its role in AD has been widely investigated. Recently, it has been reported that a circular RNA (circRNA) originated from *APP* gene can serve as a template for Aβ synthesis, postulating it as an alternative pathway for the Aβ biogenesis. Moreover, circRNAs play important roles in brain development and in neurological diseases. Therefore, our aim was to study the expression of a circAPP (hsa_circ_0007556) and its linear cognate in AD human entorhinal cortex, a brain region most vulnerable to AD pathology. First, we confirmed the presence of circAPP (hsa_circ_0007556) in human entorhinal cortex samples using RT-PCR and Sanger sequencing of PCR products. Next, a 0.49-fold decrease in circAPP (hsa_circ_0007556) levels was observed in entorhinal cortex of AD cases compared to controls (*p*-value < 0.05) by qPCR. In contrast, *APP* mRNA expression did not show changes in the entorhinal cortex between AD cases and controls (Fold-change = 1.06; *p*-value = 0.81). A negative correlation was found between Aβ deposits and circAPP (hsa_circ_0007556) and *APP* expression levels (Rho Spearman = −0.56, *p*-value < 0.001 and Rho Spearman = −0.44, *p*-values < 0.001, respectively). Finally, by using bioinformatics tools, 17 miRNAs were predicted to bind circAPP (hsa_circ_0007556), and the functional analysis predicted that they were involved in some pathways, such as the Wnt-signaling pathway (*p* = 3.32 × 10^−6^). Long-term potentiation (*p* = 2.86 × 10^−5^), among others, is known to be altered in AD. To sum up, we show that circAPP (hsa_circ_0007556) is deregulated in the entorhinal cortex of AD patients. These results add to the notion that circAPP (hsa_circ_0007556) could be playing a role in the pathogenesis of AD disease.

## 1. Introduction

Alzheimer’s disease (AD) is a chronic and irreversible neurodegenerative disease [1]. AD is the leading cause of age-related dementia and is also the most common neurodegenerative disease [2,3]. The main anatomopathological features of AD are the brain deposition of intraneuronal neurofibrillary tangles (NFTs) of hyperphosphorylated tau protein and extracellular plaques of Amyloid beta peptide (Aβ) in the parenchyma and blood vessels.

AD can be classified into two types, familial AD and sporadic AD. The former constitutes 5% of AD cases and is characterized by mutations in the Amyloid Precursor Protein (*APP*), Presenilin 1 (*PSEN1*) or Presenilin 2 (*PSEN2*) genes, all of which are related to the synthesis and processing of Aβ-peptide. Regarding sporadic AD, the cause of the disease remains unknown, although it is considered a multifactorial disease in which genetic and environmental risk factors contribute to its development [2]. 

*APP* is a membrane protein which performs several crucial cellular functions. So far, a number of *APP* isoforms have been described by alternative splicing; one of which is commonly expressed in the brain, where it participates in synaptogenesis and synaptic plasticity [4,5]. *APP* can be processed through the amyloidogenic pathway, in which the final products are going to be the Aβ40 and Aβ42 peptides, components of amyloid plaque, and the *APP* intracellular domain (amyloid precursor protein intracellular domain, AICD), following proteolytic cleavage of the *APP* protein carried out by the enzymes β-secretase (BACE1) and γ-secretase. In the non-amyloidogenic pathway, the enzymes in charge of processing the *APP* protein are α-secretase (ADAM metallopeptidase domain 10, ADAM10) and γ-secretase, and the final product will be irrelevant peptides for this pathology [1,6]. 

Although the role of Aβ in the AD pathogenesis is yet not well understood [7], and some authors believe that aberrant Aβ expression may not be the primary cause of all EOAD [8], the amyloid cascade theory still remains the most widely accepted pathogenic model [6]. This theory proposes that Aβ deposition is the first critical event that triggers a cascade of molecular phenomena leading to neurofibrillary deposits, synaptic failure, neuronal death, neurodegeneration and, finally, AD dementia. It seems clear that deposition of Aβ occurs as a consequence of altered processing of *APP* or Aβ clearance. However, it is still not well understood how the peptide is produced [9]. 

Recently, it has been described that alternative splicing of *APP* results in the generation of several circRNAs. Mo et al. even demonstrated that Aβ can be transcribed from a specific circRNA (hsa_circ_0007556) [9]. circRNAs are single-stranded RNA molecules characterized, as their name suggests, by a circular structure. They lack 5′ and 3′ ends since, after transcription, a covalent bond is established between these ends [10,11]. circRNAs can act as miRNA sponges, protein templates or transcriptional regulators, among other described functions [12,13]. circRNAs are evolutionarily conserved and expressed in a large number of body tissues [10,11,14,15]. However, it is in the brain that they are most highly expressed; in fact, 20% of brain genes encode circRNAs [16,17]. Interestingly, neuronal specialization has been related to a high level of alternative splicing [18], and this process could also explain the enormous amount of brain-specific circRNAs [14]. Nevertheless, the expression levels of circRNAs vary from one brain region to another, with the synapse being the site where their expression is highest [19]. Thus, alterations in the expression levels of circRNAs have been described in different neurological diseases, such as multiple sclerosis (MS), Parkinson’s disease (PD), amyotrophic lateral sclerosis (ALS) and AD, among others [20,21,22,23,24].

In the last few years, thanks to advances in massive sequencing and transcriptome analysis, knowledge about circRNAs involvement in AD has increased exponentially. Numerous studies, both in human brain or blood samples and in cellular and mouse models of AD, have shown dysregulation of circRNAs in this disease. A number of circRNAs are now known to play important regulatory functions in neuroinflammation (e.g., circ_0000950), oxidative stress (e.g., mmu_circRNA_013636 and mmu_circRNA_012180) and autophagy (e.g., circNF1-419), as well as Aβ production and degradation (e.g., circHDAC9, CDR1as, circHOMER1 or circCORO1C) [25,26,27,28,29,30,31,32,33]. In addition, significant changes in circRNAs expression have been detected whose genes of origin are closely related to AD pathology, involved in synaptic plasticity and neuronal survival (e.g., *HOMER1*, *DOCK1*, *NTRK2* and *APC*) or vesicular trafficking (e.g., DGL1/SAP97, TRAPPC9 and KIF1B) [34]. From the *APP* and the *MAPT* genes, which encode the main proteins involved in AD [9,35], or from the *APOE* gene [36], circRNAs also originate. Likewise, several possible candidate circRNAs for diagnosis, prognosis and disease progression have been postulated, such as circHOMER1, circCOROC1 or hsa_circ_0003391 [25,26,37,38,39,40]. Of all these, circRNAs derived from the *APP* gene are particularly interesting. Thirty-three circRNAs derived from the *APP* gene (circAPPs) have been described by in silico analysis of RNA sequencing data, and 17 circAPP have been observed by RT-PCR and Sanger sequencing. Interestingly, only two of them, namely hsa_circ_0007556 and hsa_circ_0115725, were in both lists, with hsa_circ_0007556 being the most abundant one [9,14,41,42]. It is known that this circAPP (hsa_circ_0007556) can be detected in two brain regions, hippocampus and prefrontal lobe, but its expression in the entorhinal cortex is unknown. Therefore, our aim was to study the expression of circAPP (hsa_circ_0007556) and its related linear form in human entorhinal cortex, given that knowledge of the circAPP (hsa_circ_0007556) expression pattern of in both healthy and AD-affected human entorhinal cortex is scarce, and this region constitutes one of the brain areas most vulnerable to the development of AD. Neuronal loss at entorhinal cortex occurs very early in AD [43], and dysfunction in this region is implicated in memory and learning impairment [44]. In addition, entorhinal cortex is the main communication pathway between the hippocampus and neocortex [44]. Furthermore, gene expression profiling changes related with memory and learning functions have been identified in entorhinal cortex from AD and control post-mortem brain samples [45]. 

## 2. Results

### 2.1. circAPP (hsa_circ_0007556) Identification in Entorhinal Human Brain

First, we wanted to test whether circAPP (hsa_circ_0007556) was expressed in the human entorhinal cortex. For this purpose, *APP*-specific divergent primers were designed to amplify circular but not linear RNA. Then, RT-PCR was performed on RNA samples isolated from the entorhinal cortex of AD patients and controls. After electrophoresis, PCR products were selected from the agarose gel for Sanger sequencing analysis (Figure 1). As a result, we succeeded to amplify circAPP (hsa_circ_0007556) in the human entorhinal cortex in both AD patients and controls. 

### 2.2. Differential Expression of circAPP (hsa_circ_0007556) and APP mRNA in AD Entorhinal Cortex

In order to study the expression levels of circAPP (hsa_circ_0007556) detected in the previous section and to explore whether there were expression differences in entorhinal cortex samples between AD and controls, RT-qPCR technique was performed. A total of 29 entorhinal cortex samples from AD patients and 16 controls were studied. All samples passed the RNA quality criteria. 

It should be noted that statistically significant differences in age (mean SD, 55.94 ± 5.31 in controls versus 82.07 ± 1.96 in AD, *p*-value < 0.001) and gender (% female, 31.25% in controls versus 62.07% in AD, *p*-value < 0.05) were found between AD samples and controls, so we fitted multivariate linear regression models for each transcript (circAPP (hsa_circ_0007556) and its corresponding linear mRNA), including the presence or absence of the disease as predictor and adjusting for age and gender as potential confounders, and it was found that only the presence of the disease significantly explained differences in the expression of circAPP transcript (*p* = 0.025). Regression coefficients and standard errors can be found in Appendix A. Additionally, we checked whether circAPP and *APP* mRNA expression is decreased with age (Appendix A) or by gender. We observed no correlation between circAPP expression and age (*p*-value = 0.306) or gender (*p*-value = 0.655), nor between *APP* mRNA expression and age (*p*-value = 0.158) or gender (*p*-value = 0.063). 

A significant decrease (fold-change (FC) = 0.49, *p*-value < 0.05) in the circAPP (hsa_circ_0007556) levels was observed in the entorhinal cortex of AD cases compared to controls. However, *APP* mRNA expression (FC = 0.80, *p*-value = 0.81) showed no significant changes in the entorhinal cortex region of AD samples compared to controls (Figure 2A). To evaluate whether the magnitude of the differences in the transcript expression levels between AD cases and controls was gender-dependent, we included in the multivariate linear regression models the interaction between the presence/absence of the disease and gender, resulting in non-statistically significant interactions (circAPP (hsa_circ_0007556), *p*-value = 0.544 and for *APP* mRNA, *p*-value = 0.549). These results demonstrated the absence of sex dependence in the differential expression levels of the transcripts between AD cases and controls.

We also wanted to analyze whether circAPP (hsa_circ_0007556) and *APP* mRNA expression changed across AD neuropathological stages, according to the ABC score. A significant decrease in circAPP (hsa_circ_0007556) expression was observed in the group with higher ABC score values with respect to controls (*p*-value < 0.05) (Figure 2B). For its part, *APP* linear transcript showed a significant decrease in expression in the group with high level of ABC score with respect to the intermediate group (*p*-value < 0.05) and the low group (*p*-value < 0.001) (Figure 2B).

Next, using a univariate general linear model, where the dependent variable was the RNA expression levels and the independent variables were the specific RNA variants (circAPP (hsa_circ_0007556) and *APP* mRNA) and the presence of disease (AD or control), it was observed that the expression of the two transcripts together in AD cases was 37.75% lower (*p*-values < 0.05) with respect to the controls. Considering linear *APP* as a reference, circAPP (hsa_circ_0007556) expression was 99.93% lower (*p*-value < 0.0001). However, when the cases (AD and control) and the two RNA variants were studied together in the model, relative expression ratios of the two transcripts (circAPP/*APP*) were maintained in the AD cases and controls (99.94% and 99.90% lower circAPP (hsa_circ_0007556) relative to *APP* mRNA, respectively, *p*-value = 0.100), meaning that the decrease in expression of the 2 RNA variants is proportional within AD cases, although only circAPP (hsa_circ_0007556) was significantly decreased in samples with AD compared to controls (Appendix A). 

We repeated the above analysis, but taking into consideration the ABC score instead of the presence of the disease (AD or control). Relative expression ratios of the two transcripts (circAPP/*APP*) were also maintained in each level of ABC scores and controls (99.95% at the low level, 99.93% at the intermediate level, 99.93% at the high level, 99.90% at the control level and lower circAPP relative to *APP* mRNA, *p*-value = 0.455, *p*-value = 0.955, *p*-value = 0.362 between groups) (Appendix A).

We can conclude that the decrease in expression of the different variants is proportional within AD or ABC score levels and control samples; in fact, the circAPP/*APP* ratio is maintained between AD patients or ABC score levels and controls.

### 2.3. Correlation between circAPP (hsa_circ_0007556) and APP mRNA Expression and Aβ Deposits

Since one of the main pathophysiological features of AD is the deposition of Aβ peptide, which derives from *APP* protein processing, we sought to study the relationship between quantitative assessment of Aβ deposits and *APP*-derived RNA transcripts expression levels. Thus, a negative correlation was found between Aβ deposits and both circAPP (hsa_circ_0007556) and linear *APP* expression (Rho Spearman = −0.48, *p*-value < 0.05 and Rho Spearman = −0.652, *p*-value <0.01, respectively) (Table 1, Appendix A).

Additionally, we decided to study the correlation between circAPP (hsa_circ_007556) or *APP* mRNA expression and the ABC score and each group individually forming ABC score (Method of Thal, Braak and Braak classification, Method of CERAD). We observed a negative correlation between circAPP (hsa_circ_007556) expression and all pathological features analyzed (Method of Thal, Braak and Braak classification, Method of CERAD, ABC score and Global average area of Aβ deposits), while *APP* mRNA showed a negative correlation with all of them, except the Method of CERAD) (Table 1).

### 2.4. In Silico Prediction of Biological Function

It has been demonstrated that circAPP (hsa_circ_0007556) could be the template for Aβ peptide transcription, but little is known about the role of circAPP (hsa_circ_0007556) in the brain [9]. Since circRNAs may function as microRNAs sponges, we decided to identify those miRNAs that target circAPP (hsa_circ_0007556). For this purpose, we used the miRNA target sites tool from CircInteractome database [46]. We identified 17 miRNAs that could potentially target circAPP (hsa_circ_0007556) (Table 2). Some of these miRNAs have been described in different neurological disorders, such as major depression, epilepsy, ALS, PD or AD [47,48,49,50,51,52,53,54]. Moreover, we wanted to know the biological pathways in which these miRNAs are involve and tried to approach the biological function of circAPP (hsa_circ_0007556). We employed the microT-CDS tool of the DIANA mirPath v.3 software [55] and used the 17 miRNAs as the input. A significant association between these miRNAs and diverse KEGG pathways were found, as follows: Wnt-signaling pathway (*p* = 3.32 × 10^−6^), Long-term potentiation (*p* = 2.86 × 10^−5^), Glycosaminoglycan biosynthesis—heparan sulfate/heparin (hsa00534) (*p* = 3.38 × 10^−5^), Ubiquitin mediated proteolysis (*p* = 6.68 × 10^−5^), Axon guidance (7.01 × 10^−5^) and Glutamatergic synapse (*p* = 7.76 × 10^−5^), among others (Appendix A).

## 3. Discussion

In this study, we observed downregulation of circAPP (hsa_circ_0007556) expression in the entorhinal cortex of AD patients versus controls. Furthermore, taking into account the progression of AD neuropathological change and according to the ABC score, circAPP (hsa_circ_0007556) expression was downregulated in the group with higher ABC score values with respect to controls and *APP* mRNA showed downregulation in the group with higher ABC score values with respect to intermediate and low groups. In addition, both transcripts showed a negative correlation with Aβ deposits in the brain tissue.

The *APP* protein is the precursor of Aβ peptides and its role in AD has been extensively investigated [73]. Databases of circRNAs predict up to 33 different circRNAs derived from the *APP* gene. Some of these circRNAs were detected by massive sequencing in different brain regions such as cerebellum, frontal cortex, diencephalon, occipital lobe, parietal lobe and temporal lobe [14,41,42]. In recent years, the focus has been on studying the expression of circRNAs in different brain regions affected by AD. For example, Lo et al. [40] detected by RNA-seq, in four brain regions, i.e., the anterior prefrontal cortex, superior temporal lobe, inferior frontal lobe and hippocampus, five circRNAs originating from *APP*, but only one was differentially expressed in the inferior frontal lobe between AD patients and controls. However, they did not observe the circAPP (hsa_circ_0007556) studied in the present work. On the other hand, Dube et al. [25] surveyed the parietal cortex, inferior frontal lobe, frontal pole, superior temporal lobe and parahippocampal lobe also by RNA-seq but found none of the *APP*-derived circRNAs. That could be because they may not meet the minimum requirements for the subsequent differential analysis.

It is worth noting that, of all the circAPPs revealed by massive sequencing, only hsa_circ_0007556 and hsa_circ_0115725 has been validated by another technique in two brain regions, the hippocampus and prefrontal lobe [9]. Following the results obtained in our study, the expression of hsa_circ_0007556 in the entorhinal cortex of both AD patients and controls can also be confirmed.

Moreover, with this work, knowledge of the differential expression of circAPP (hsa_circ_0007556), which was found to be less expressed in samples with AD compared to controls, is added. A priori, an increase in its expression would have been expected, since the mechanism by which Aβ deposition is increased in AD brain is not entirely clear, and an increase in this circRNA could imply an increase in the production of Aβ [9]. In fact, it has been recently published that this circAPP (hsa_circ_0007556) may serve as a “template” for Aβ synthesis, postulating itself as an alternative pathway for this peptide biogenesis [9]. On the other hand, a negative correlation of peptide Aβ burden and circAPP (hsa_circ_0007556) expression levels is observed here (although when was taking into consideration the ABC score, only the group with higher ABC score values showed a significant decrease in circAPP (hsa_circ_0007556) expression respect to control, but this observation could be due to the reduced sample size by segmenting into ABC stages limits the power of the analysis), leading to hypothesize that it could play a regulatory role, directly or indirectly, on the expression of enzymes in charge of *APP* protein processing. For example, due to the upregulation of the peptide, a downregulation of circRNA could be induced to compensate for the excess of Aβ that could be produced from it. In fact, the expression of this circRNA seems to be reduced across the progression of the disease. An alternative explanation would be that enzymes involved in Aβ peptide processing may bind to circAPP (hsa_circ_0007556), so that circAPP (hsa_circ_0007556) would control the availability of them; a circAPP (hsa_circ_0007556) dysfunction would lead to the release of the enzymes, resulting in an increase of Aβ. In any case, our study is observational and merely shows a statistical association, not causality. Therefore, to elucidate whether there is a true involvement of circAPP (hsa_circ_0007556) in the generation of the peptide, further experimental studies would be necessary.

In any case, it would be interesting to study several aspects of this circRNA, such as the expression of the protein that originates from circAPP (hsa_circ_0007556), the methylation of circAPP (hsa_circ_0007556), since it has been demonstrated that m^6^A RNA methylation can efficiently promote translation to proteins [27] or the interaction of this circRNA with other molecules to try to elucidate its function in the brain and its role in AD. It would also be interesting to study the localization of this circRNA at the cellular level, as it is known that the expression of circRNAs is enriched at the synapse [16], and synaptic dysfunction is one of the characteristics of AD [7].

Anyway, it must be taken into consideration that the decrease in the level of circAPP may be due to a number of reasons, such as downregulation of its expression, higher processing or destruction or sequestration of circRNA in the amyloid plaques.

Although circAPP (hsa_circ_0007556) expression is downregulated, the expression of linear *APP* is not altered on the whole. These findings are in the same direction as others previously shown in the literature [9]. Nevertheless, one would expect that the ratio of circAPP (hsa_circ_0007556) to linear *APP* between patients and controls may be altered, given that circAPP (hsa_circ_0007556) shows expression changes in the face of AD while *APP* does not. However, what is observed is that the circAPP (hsa_circ_0007556)/*APP* ratio is maintained between AD patients and controls. This may be due to a decrease in *APP* expression, although not statistically significant, in AD patients and this small decrease may be sufficient to maintain the circAPP (hsa_circ_0007556)/*APP* ratio.

Both RNA transcripts show a negative correlation with Aβ deposits, and this points to other mechanisms being involved in this process, such as those concerning miRNAs. In the CircInteractome database [46], circAPP (hsa_circ_0007556) is predicted to have a binding site for 17 miRNAs. Some of them have been associated with several diseases, but it is worth mentioning that at least five of them have been related with AD. For example, hsa-miR-598, hsa-miRNA-659 and hsa-miR-324-5p have been identified as candidate biomarkers in different fluids such as plasma or cerebrospinal fluid [47,48,49]. In an AD model in primary mouse hippocampal neurons, circ_0004381 was found to regulate *PSEN1* expression through miR-647 [67]. has-miR-186 is a strong negative regulator of *BACE1* expression, a protein involved in the amyloidogenic pathway [61]. Furthermore, in the review by He et al. [74], it is pointed out that different miRNAs participate in the metabolism of Aβ peptides acting at different levels and on different genes.

Moreover, these 17 miRNAs found in the in silico functional analysis have been significantly associated with multiple pathways, some of which are altered in AD. For instance, Wnt-signaling pathway is implicated in neurodevelopment and neurogenesis and, in AD, is downregulated in several cell types in AD brains [75]. Other relevant pathways, such as synaptic long-term potentiation, axon guidance and dysregulation in ubiquitin-mediated proteolysis, which are implicated in synaptic plasticity and synaptic development, have been related to cognitive impairment observed in AD [76,77,78]. Glycosaminoglycan biosynthesis—heparan sulfate/heparin is associated with the formation of Aβ plaques [79,80,81], and the glutamatergic synapse pathway has been observed altered in AD, while glutamatergic receptors have been suggested as pharmacological targets [82]. However, this is still an in silico study, and other functional studies are required to confirm these predictions.

## 4. Methods

### 4.1. Human Entorhinal Cortex Samples

Brain entorhinal cortex samples from 29 AD patients and 16 controls were provided by Navarrabiomed Brain Bank. After death, half-brain specimens from donors were cryopreserved at −80 °C. A neuropathological examination was completed following the usual recommendations [83] and according to the updated National Institute on Aging-Alzheimer’s Association guidelines [84].

Assessment of Aβ deposition was carried out by immunohistochemical staining of paraffin-embedded sections (3–5 μm thick) with a mouse monoclonal (S6 F/3D) anti-Aβ antibody (Leica Biosystems Newcastle Ltd., Newcastle upon Tyne, United Kingdom). Evaluation of neurofibrillary pathology was performed with a mouse monoclonal antibody anti-human PHF-TAU, clone AT-8 (Tau AT8) (Innogenetics, Gent, Belgium), which identifies hyperphosphorylated tau (*p*-tau) [85]. The reaction product was visualized using an automated slide immunostainer (Leica Bond Max) with Bond Polymer Refine Detection (Leica Biosystems, Newcastle Ltd., UK). Other protein deposits, such as synuclein deposits, were ruled out by a monoclonal antibody against α-synuclein (NCL-L-ASYN; Leica Biosystems, Wetzlar, Germany). The staging of AD was performed by using the ABC score according to the updated National Institute on Aging-Alzheimer’s Association guidelines [84]. ABC score combines histopathologic assessments of Aβ deposits determined by the method of Thal (A) [84], staging of neurofibrillary tangles by Braak and Braak classification (B) [85], and scoring of neuritic plaques by the method of CERAD (Consortium to Establish A Registry for Alzheimer’s Disease) (C) [86] to characterize AD neuropathological changes. Thus, the ABC score shows three levels of AD neuropathological severity: low, intermediate and high. A summary of the characteristics of subjects considered in this study is shown in Appendix A.

### 4.2. RNA Isolation and Reverse Transcription—Polymerase Chain Reaction (RT-PCR)

Total RNA, including small RNA species, was isolated from cells and entorhinal cortex samples with miRNAeasy mini Kit (QIAGEN, Redwood City, CA, USA) following the manufacturer’s instructions. Concentration and purity of RNA were both evaluated with NanoDrop spectrophotometer. Complementary DNA (cDNA) was reverse transcribed from 500 ng total RNA with SuperScript^®^ III First-Strand Synthesis Reverse Transcriptase (Invitrogen, Carlsbad, CA, USA) after priming with random primers. RT-PCR was performed by using GoTaq^®^ DNA polymerase (Promega, Madison, WI, USA) in an Applied Biosystems™ Veriti™ Thermal Cycler, 96-Well (Applied Biosystems, Foster City, CA, USA). PCR conditions were as follows: denaturation at 95 °C for 20 s, extension at 72 °C for 30 s and annealing temperatures 60 °C for 40s and cycles used were 40. Primer3 software was used for divergent primers design (Appendix A).

### 4.3. Candidate Band Selection and Sanger Sequencing

Candidate bands were selected after 1.8% agarose gel electrophoresis of RT-PCR products. Bands purification were made with Wizard^®^ SV Gel and PCR Clean-Up System (Promega, Madison, WI, USA). Next, Sanger sequencing was performed and UCSC (University of California Santa Cruz) Genome Browser software was used for the sequence alignment [87,88].

### 4.4. Real Time Quantitative PCR (RT-qPCR) Assay

Total RNA were isolated from the entorhinal cortex with RNAeasy Lipid Tissue mini Kit (QIAGEN, Redwood City, CA, USA) following the manufacturer’s instructions. Genomic DNA was removed with recombinant DNase (TURBO DNA-free™ Kit, Ambion, Austin, TX, USA). Concentration and purity of RNA were both evaluated with NanoDrop spectrophotometer. Complementary DNA (cDNA) was reverse transcribed from 500 ng total RNA with SuperScript^®^ III First-Strand Synthesis Reverse Transcriptase (Invitrogen, Carlsbad, CA, USA) after priming with random primers. RT-qPCR reactions were performed in triplicate with Power SYBR Green PCR Master Mix (Invitrogen, Carlsbad, CA, USA) on the QuantStudio 12K Flex real-time PCR system (Applied Biosystems, Foster City, CA, USA) and repeated twice on independent cDNA samples. The sequences of convergent primer pairs for linear RNA detection were designed using the IDT real-time PCR tool (Coralville, IA, USA) and Primer3 software and are listed in Appendix A. The relative expression level of mRNA in each sample was calculated using the delta delta-CT method and the geometric mean of *GAPDH* and *ACTB* genes was used as a reference to normalize the expression values [89].

### 4.5. Quantitative Assessment of Aβ Deposits in Brain Tissues

In order to quantitatively assess the Aβ burden for further statistical analysis, we applied a method to quantify protein deposits. This method generates a numeric measurement that represents the extent of Aβ deposition. Sections of the entorhinal cortex were examined after performing immunostaining with anti Aβ antibody as described above in Human Entorhinal Samples. Three pictures were obtained for each immunostained section by using an Olympus BX51 microscope at ×10 magnification power. Focal deposit of Aβ, as described by Braak & Braak (neuritic, immature, and compact plaque) [85], was manually determined and was further edited and analyzed with ImageJ software. Then, the Aβ plaque count, referred to as amyloid plaque score (APS) and total area of Aβ deposition, was automatically measured by ImageJ and averaged for each section (Appendix A).

### 4.6. Prediction of circAPP (hsa_circ_0007556) Interaction with miRNAs and Biological Function

With the help of public databases containing information about ncRNAs we can predict which genes, proteins and miRNAs candidate circRNAs interact with. For the prediction of miRNAs that could bind to circAPP (hsa_circ_0007556), the miRNA target sites tool from the CircInteractome database was used [46]. For the study of the potential biological pathways that miRNAs might be regulating, microT-CDS tool of the DIANA mirPath v.3 software [55] was employed, and as for input, we included the miRNAs obtained in the previous step.

### 4.7. Statistical Analysis

Statistical analysis was performed with SPSS 25.0 (IBM, Inc., Chicago, IL, USA). Before performing differential analysis, we checked whether continuous variables follow a normal distribution, as per one-sample Kolgomorov–Smirnov test and the normal quantil–quantil (Q–Q) plots. For the analysis of the differential expression of the distinct *APP* transcripts, we fitted multivariate linear regression models for each transcript (circAPP and its corresponding linear mRNA), including the presence or absence of the disease as predictor and adjusting for age and gender as potential confounders. All three explanatory variables were included using the Enter method (all variables are entered in a single step). To evaluate the homogeneity of variances, the Levene’s test was used, and the normality of the regression residuals was assessed by visualization of the histograms and Q–Q plots. All models met the aforementioned requirements. In order to analyze differences in the expression levels of the different transcripts studied between the ABC scale groups, a general univariate linear model adjusted for gender and age and the Bonferroni post hoc test were developed. On the other hand, a general linear univariate model was used to determine the proportions between the expression levels of the *APP* RNA variants in the AD samples versus the control samples and ABC score stages. For each gene, an expression variable was created where the log(expression) of each variant and another categorical variable with the type of transcript was collected, leaving a model where the expression variable was the dependent variable and as fixed factors the case variables ((AD or control) or (ABC score stages (control, low, intermediate and high) and the type of transcript were included. Spearman’s test was used to assess the correlation between the continuous variables circRNA or mRNA expression and Aβ deposition or Method of Thal or Braak and Braak classification or Method of CERAD or ABC score. GraphPad Prism version 9.00 for Windows (GraphPad Software, La Jolla, CA, USA) was used to draw graphs.

## 5. Conclusions

We observed the expression of circAPP (hsa_circ_0007556) in the human entorhinal cortex, and we also show that circAPP (hsa_circ_0007556) is downregulated in the entorhinal cortex of AD patients compared to controls and at late stages of neuropathological changes. These results add to the notion that circAPP (hsa_circ_0007556) could be playing a role in the pathogenesis of AD disease.

## Figures and Tables

**Figure 1 ijms-24-04308-f001:**
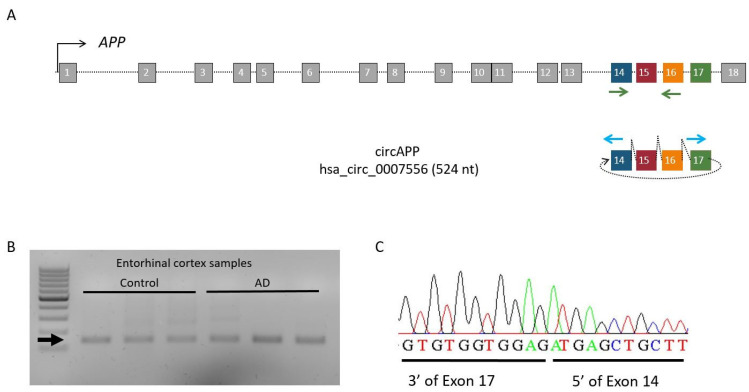
Identification of circAPP (hsa_circ_0007556). (**A**) Schematic representation of circAPP (hsa_circ_0007556) and *APP* mRNA, arrows represent primers (blue arrows for circAPP (hsa_circ_0007556) divergent and green arrows for *APP* mRNA). (**B**) Electrophoresis agarose gel of circAPP (hsa_circ_0007556), arrow points the circAPP (hsa_circ_0007556) PCR product. (**C**) Electropherogram showing the circAPP (hsa_circ_0007556) covalent junction.

**Figure 2 ijms-24-04308-f002:**
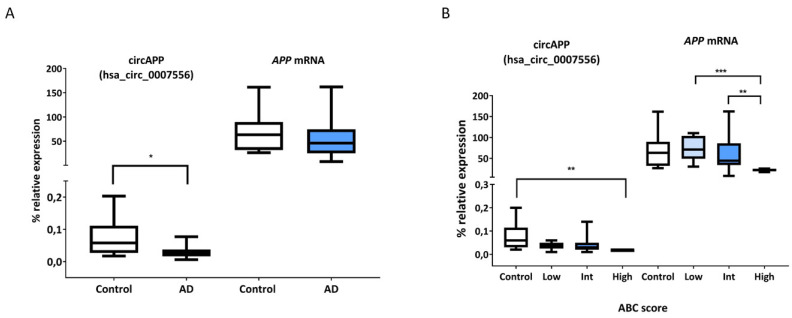
Expression of circAPP (hsa_circ_0007556) and *APP* mRNA and as a function of the ABC score. (**A**) The box-plot represents the percentage of relative expression of circAPP (hsa_circ_0007556) and *APP* mRNA in patients and controls with respect to the geometric mean of the housekeeping genes *GAPDH* and *ACTB*. (**B**) The box-plot represents the percentage of the relative expression of circAPP (hsa_circ_0007556) and *APP* mRNA according to the ABC score groups, with respect to the geometric mean of the housekeeping genes *GAPDH* and *ACTB*. * *p*-value < 0.05, ** *p*-value < 0.01 and *** *p*-value < 0.001.

**Table 1 ijms-24-04308-t001:** Correlation between the Method of Thal, Braak and Braak classification, Method of CERAD, Global average area of Aβ deposits and circAPP or *APP* mRNA expression levels.

		Method of Thal	Braak and Braak Classification	Method of CERAD	ABC Score	Global Average Area of Aβ Deposits
log(circAPP)	Correlation coeficient (Spearman’s Rho)	−0.496 **	−0.503 **	−0.499 **	−0.515 **	−0.480 *
	Sig. (bilateral)	0.001	0.001	0.001	0.000	0.024
	N	43	44	43	43	22
log(*APP* mRNA)	Correlation coeficient (Spearman’s Rho)	−0.339 *	−0.415 **	−0.243	−0.384 *	−0.652 **
	Sig. (bilateral)	0.026	0.005	0.116	0.011	0.001
	N	43	44	43	43	22

* *p*-value < 0.05, ** *p*-value < 0.01.

**Table 2 ijms-24-04308-t002:** miRNAs predicted to bind circAPP (hsa_circ_0007556) with circInteractome tool and their implication in neurological disorders.

miRNA	Neurological Disorders Related	Expression in AD	Tissue/Fluid Studied	References
hsa-miR-1200	-			
hsa-miR-1208	-			
hsa-miR-1270	-			
hsa-miR-1272	-			
hsa-miR-136	Prion disease, epilepsy, depression, PD			[50,51,53,56]
hsa-miR-186	AD, ischemic stroke	-	Neuro-2a cells, 7PA2 cells	[57,58,59,60,61]
hsa-miR-324-5p	AD	↓	plasma	[49,62]
hsa-miR-421	PD, epilepsy, ischemic stroke			[63,64,65,66]
hsa-miR-518a-5p	-			
hsa-miR-527	-			
hsa-miR-549	-			
hsa-miR-598	AD	No detected	CSF	[48]
		↑(tendency)	CSF exosomes	
hsa-miR-620	-			
hsa-miR-646	-			
hsa-miR-647	AD			[67]
hsa-miR-659	AD, ischemic stroke, frontotemporal dementia, MS	↓	hippocampal neurons AD mouse model	[47,68,69,70,71]
hsa-miR-876-3p	Epilepsy	↑	Plasma exosomes	[72]

## Data Availability

Not applicable.

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
