# Peer review of "circRNA from *APP* Gene Changes in Alzheimer’s Disease Human Brain"

_ijms, 2023, doi:10.3390/ijms24054308_

Round 1

Reviewer 1 Report

This manuscript by Amaya Urdánoz-Casado and colleagues showed decrease of circAPP expression in the entorhinal cortex of AD patients, and possible involvement of circAPP in the Alzheimer’s disease. Although the idea that circRNA derived from the APP gene involves in the AD development is potentially interesting, I do not think that the present set of data are conclusive enough to draw unambiguous structural conclusions.

Major points

1)     The authors investigate the expression of circAPP and APP mRNA in the entorhinal cortex of AD patients and healthy control because this region is most vulnerable to the development of AD (line110-). This sentence is not sufficient to explain the importance of the entorhinal cortex in AD development.

2)     As long as I know, the entorhinal cortex is one of the first region affected by AD and most damaged area in AD. However, the significant decrease in circAPP expression can be detected only at high ABC score (Fig.2). This means that circAPP may not be involved in the AD pathogenesis, just a consequence.

3)     The age of healthy control and AD samples is way different as the authors said. The authors explained that the differential circAPP expression was adjust with univariate linear model (line 141). This sentence is not sufficient for explaining the experiments and analyses. The authors should explain this method somewhere in the Method section or Supplementary information. In addition, the authors need to show whether circAPP expression is decreased with the age.

4)     Since one of the amyloid plaque components is RNA aptamer, the size of which is less than 100nt, it is possible that circAPP expression is relatively lower in the samples with more amyloid plaques. In fact, both expression of circAPP and APP mRNA was decreased with ABC score severity (Fig.2). To solve this problem, I recommend to show the expression of another type of circAPP (circ-0115725) or other RNA as a control.

5)     The authors listed some of the miRNAs as a target of circAPP (Table 1). The authors had better show the expression of these miRNAs in the entorhinal cortex and compare the differential expression between AD and HC samples.

6)    In addition, the authors listed several target proteins or pathways of circAPP in the discussion.  To reinforce the argument, the comprehensive analysis using AD entorhinal cortex is definitely needed.

Minor Points

1)     It is very hard to see the band in electrophoresis agarose gel (Fig.1b). In addition, there seems to be no difference between AD and HC samples. I recommend performing the same experiment again and replace it.

Author Response

Response to the Reviewers’ Comments

We are very grateful of the reviewers’ effort in revising the manuscript and their constructive suggestions.

Review 1

This manuscript by Amaya Urdánoz-Casado and colleagues showed decrease of circAPP expression in the entorhinal cortex of AD patients, and possible involvement of circAPP in the Alzheimer’s disease. Although the idea that circRNA derived from the APP gene involves in the AD development is potentially interesting, I do not think that the present set of data are conclusive enough to draw unambiguous structural conclusions.

Major points

1)     The authors investigate the expression of circAPP and APP mRNA in the entorhinal cortex of AD patients and healthy control because this region is most vulnerable to the development of AD (line110-). This sentence is not sufficient to explain the importance of the entorhinal cortex in AD development.

We add the following paragraph to the main text:

‘Neuronal loss at entorhinal cortex occurs very early in AD (43), and dysfunction in this region is implicated in memory and learning impairment (44). In addition, entorhinal cortex is the main communication pathway between the hippocampus and neocortex (44). Furthermore, gene expression profiling changes related with memory and learning functions have been identified in entorhinal cortex from AD and control post-mortem brain samples (45).’ (Lines 114-120)

2)     As long as I know, the entorhinal cortex is one of the first region affected by AD and most damaged area in AD. However, the significant decrease in circAPP expression can be detected only at high ABC score (Fig.2). This means that circAPP may not be involved in the AD pathogenesis, just a consequence.

Thank you for your observation, it is true that the significant decrease of circAPP expression is only detected in controls compared to high ABC score group when performing contrast tests in pairs.  However, Figure 2 shows a decrease in circAPP expression levels also in the low and intermediate ABC score groups with respect to the control. Therefore, we have performed trend analysis and correlation analysis between circAPP expression and ABC score. We can observe that the trend analysis is significant (p-value<0.01) and there is an inverse correlation between circAPP expression levels and ABC score (rho spearman=-0.515. p-value<0.001). In any case, this study is observational and we can only conclude about associations, not causality, and the results of the stepwise sub-analysis raise new hypotheses (not conclusions) about causality. Other consideration is that the reduced sample size by segmenting into ABC stages limits the power of the analysis. This limitation is now included in the manuscript in lines 280-284

3)     The age of healthy control and AD samples is way different as the authors said. The authors explained that the differential circAPP expression was adjust with univariate linear model (line 141). This sentence is not sufficient for explaining the experiments and analyses. The authors should explain this method somewhere in the Method section or Supplementary information. In addition, the authors need to show whether circAPP expression is decreased with the age.

We have improved the explanation of the analyses carried out in both results (lines 146-151) and the Methods section (lines 430-437).

As suggested by the Reviewer, we have checked whether circAPP and APP mRNA expression is decreased with age, and we can observe that there is no correlation between circAPP expression and age, or between APP mRNA expression and age. We have add it in the manuscript (Lines 151-155, Figure S1)

4)     Since one of the amyloid plaque components is RNA aptamer, the size of which is less than 100nt, it is possible that circAPP expression is relatively lower in the samples with more amyloid plaques. In fact, both expression of circAPP and APP mRNA was decreased with ABC score severity (Fig.2). To solve this problem, I recommend to show the expression of another type of circAPP (circ-0115725) or other RNA as a control.

Thank you for the comment, as it provides an additional and highly interesting explanation for the decrease in circAPP levels that we observed in the EC of AD patients. We have now enriched the discussion section with the following sentence:

“Anyway, it must be taken into consideration that the decrease in the level of circAPP may be due to a number of reasons, such as downregulation of its expression, higher processing or destruction, or sequestration of circRNA in the amyloid plaques”.(lines 306-308)

However, it is still a hypothesis. And since aptamer binding is very specific, we believe that measuring the levels of another circular RNA would not give us the answer, since if it remained stable it could be because that RNA is not downregulated or does not bind to an amyloid plaque aptamer. Although this study would be beyond the scope of this article, we will keep it in mind for future experiments.

5)     The authors listed some of the miRNAs as a target of circAPP (Table 1). The authors had better show the expression of these miRNAs in the entorhinal cortex and compare the differential expression between AD and HC samples.

This is a very appealing idea to be explored. Indeed, we would also be interested in studying the profile of these miRNAs in AD samples relative to controls. However, this is outside the scope of the study this time. As we only aimed to perform an in silico study (to explore the potential functional impact of our results), we did not consider to check the expression of these miRNAs in entorhinal cortex. However, we have added the expression level of these miRNAs known to be related to AD in the table 2.

Anyway, indeed, it would be interesting to study the expression of these miRNAs in the entorhinal cortex, even more, perform a luciferase assay to determine whether these miRNAs directly target the circAPP. Thank you for the suggestion.

6)    In addition, the authors listed several target proteins or pathways of circAPP in the discussion.  To reinforce the argument, the comprehensive analysis using AD entorhinal cortex is definitely needed.

In order to know the signaling pathways in which circAPP is involved, such as the molecules it interacts with, a comprehensive analysis would be necessary, in order to elucidate the role of circAPP in AD. However, as mentioned above, our intention was to confirm and quantify circAPP expression in the entorhinal cortex for the first time, to open the door to in-depth study of circAPP.

Minor Points

1)     It is very hard to see the band in electrophoresis agarose gel (Fig.1b). In addition, there seems to be no difference between AD and HC samples. I recommend performing the same experiment again and replace it.

Thanks for the suggestions, we have repeated the experiment and the electrophoresis agarose gel now shows band of higher intensity improving the quality of the Figure 1.

We certainly agree with the reviewer that there seems to be no difference between AD and HC samples when looking only at the image. Honestly, the main purpose of this experiment was to confirm that circAPP is expressed in AD and in control samples. But, in this regard, we decided to perform RT-qPCR to better reflect differences of the relative expression level of circAPP by using a housekeeping gene as reference to normalize the expression levels.

Reviewer 2 Report

Urdanoz-Casado and colleagues examine expression of a circular RNA from the APP gene in AD cases and healthy controls. The circRNA they examine has potential to contribute to Abeta peptide production, and they correlate circRNA expression with Abeta deposits as well as overall AD pathology score (ABC score). They surprisingly find reduced circAPP levels (but not APP mRNA levels) in AD patients vs. controls, as well as in patients with high ABC score. They also find negative correlations between circAPP and APP mRNA levels versus Abeta deposits, and speculate on functions of circAPP in disease. The experiments are straight forward, and are conducted and explained appropriately. However, the impact of the findings is limited because the functions of circAPP and any role it might play in AD are very poorly understood. Nonetheless, the results presented here do contribute to understanding how circRNAs may be regulated in disease, and could point to important regulatory functions for circAPP in neurons. I have a number of concerns that should be addressed before I can recommend publication of the manuscript.

1)    Figure 2A – the text states that APP mRNA levels in AD are 1.06-fold of levels in control samples, however the 2A plot seems to show slightly lower APP mRNA levels in AD. In the discussion the authors also mention that there is a small decrease in APP expression in AD (lines 274-276). Can the authors check these data and clarify what the actual results are?

2)    It would improve confidence in the results if the authors used second primer pairs to look at levels of circAPP and APP mRNA presented in Fig 2.

3)    The authors show decreased circAPP but unchanged APP mRNA levels in all AD cases vs. controls (Fig 2A), but then show that both exhibit reduced levels in high ABC score patients. In the discussion they mention that there is not a significant reduction in the ratio of circAPP/APP in AD patients. This lack of significance is a part of the results of the manuscript and should be included in this section.

4)    Can the authors also indicate whether circAPP/APP ratio is changed in patients with different ABC scores (2B)?

5)    How was the Thal assessment and ABC staging done for the patients/samples where Abeta plaque area was not available (according to Table S2)?  

6)    The authors show that circAPP and APP mRNA are negatively correlated with Abeta deposits. I think it would be important for the authors to provide correlations (or lack of) for all other pathological (and if possible cognitive) data they have access to. This should be easy to perform for Braak stage, neuritic plaques/CERAD, ABC score, etc, and could be included as a table in the manuscript.

7)    Relatedly, can the authors examine whether there are circAPP and/or APP correlations with traits such as age, gender and PMI?

8)    Lines 146-150 – this sentence is poorly worded and should be altered to be clearer.

9)    The authors use the term ‘lustrum’ on line 86. Despite being a native English speaker, I had to look the word up. I think a different wording here would be easier to follow, and beyond that, it appears that a number of the references referred to are more than 5 years old (so lustrum is not actually an accurate descriptor).

10) Synaptic is mis-spelled ‘synapsis’ 3 times on lines 295-297.

Author Response

Response to the Reviewers’ Comments

We are very grateful of the reviewers’ effort in revising the manuscript and their constructive suggestions.

Urdanoz-Casado and colleagues examine expression of a circular RNA from the APP gene in AD cases and healthy controls. The circRNA they examine has potential to contribute to Abeta peptide production, and they correlate circRNA expression with Abeta deposits as well as overall AD pathology score (ABC score). They surprisingly find reduced circAPP levels (but not APP mRNA levels) in AD patients vs. controls, as well as in patients with high ABC score. They also find negative correlations between circAPP and APP mRNA levels versus Abeta deposits, and speculate on functions of circAPP in disease. The experiments are straight forward, and are conducted and explained appropriately. However, the impact of the findings is limited because the functions of circAPP and any role it might play in AD are very poorly understood. Nonetheless, the results presented here do contribute to understanding how circRNAs may be regulated in disease, and could point to important regulatory functions for circAPP in neurons. I have a number of concerns that should be addressed before I can recommend publication of the manuscript.

  1. Figure 2A – the text states that APP mRNA levels in AD are 1.06-fold of levels in control samples, however the 2A plot seems to show slightly lower APP mRNA levels in AD. In the discussion the authors also mention that there is a small decrease in APP expression in AD (lines 274-276). Can the authors check these data and clarify what the actual results are?

Thank you for your observation. The FC=1.06 is indeed a mistake, we have corrected it (line 158). There is no differences in APP mRNA expression between AD and control samples (FC = 0.80, p-value = 0.81), but in Figure 2A and in Figure 2B we can observed the slight decreased expression in APP mRNA in AD samples respect to control.

  1. It would improve confidence in the results if the authors used second primer pairs to look at levels of circAPP and APP mRNA presented in Fig 2.

We totally agree with the reviewer, the use of a second primer pair would improve the confidence in the results. However, we have not considered this as the primers used showed good performance in RTqPCR for both circAPP and APP mRNA. As a general approach, we performed agarose gel electrophoresis of the RTqPCR products to confirm that the amplification is specific, we also performed the RT-qPCR reactions in triplicate and repeated twice on independent cDNA samples, to have confidence in the results. However, we will keep this in mind for future studies.

  1. The authors show decreased circAPP but unchanged APP mRNA levels in all AD cases vs. controls (Fig 2A), but then show that both exhibit reduced levels in high ABC score patients. In the discussion they mention that there is not a significant reduction in the ratio of circAPP/APP in AD patients. This lack of significance is a part of the results of the manuscript and should be included in this section.

Actually, in line 182-194 we described the ratios and the significance, but even so, we have clarified and emphasized these results in lines 201-203.

  1. Can the authors also indicate whether circAPP/APP ratio is changed in patients with different ABC scores (2B)?

Thanks for this comment as it significantly improves our results. We have added this analysis in section 2.2 Differential expression of circAPP (hsa_circ_0007556) and APP mRNA in AD entorhinal cortex (lines 195-200)

  1. How was the Thal assessment and ABC staging done for the patients/samples where Abeta plaque area was not available (according to Table S2)?  

We have measured Ab plaque areas as we describe in section 4.5. Quantitative assessment of Ab deposits in brain tissues, 6 samples do not have assessment due to the fact that we did not have access to the images. However, in all of them ThaI assessment was performed by Biobank of Navarrabiomed following the usual recommendations  and according to the updated National Institute on Aging Alzheimer’s Association guidelines  as we described in section 4.1. Human Entorhinal Cortex Samples

  1. The authors show that circAPP and APP mRNA are negatively correlated with Abeta deposits. I think it would be important for the authors to provide correlations (or lack of) for all other pathological (and if possible cognitive) data they have access to. This should be easy to perform for Braak stage, neuritic plaques/CERAD, ABC score, etc, and could be included as a table in the manuscript.

Thank you for the suggestion. We have analyzed correlation for Thal assessment, Braak stage, neuritic plaques/CERAD and ABC score and we have added these result as a table in the manuscript (Table 1) (lines 213-220), showing that circAPP expression correlates negatively with Method of Thal, Braak and Braak classification, Method of CERAD, ABC score and Global average area of Ab deposits, while, APP mRNA show negative correlation with all of them except Method of CERAD).

  1. Relatedly, can the authors examine whether there are circAPP and/or APP correlations with traits such as age, gender and PMI?

We have conducted this analysis and there is no correlation.

Age

PMI

Gender

logcircAPP

correlation coeficient (Spearman’s Rho)

-0.156

0.104

0.068

Sig. (bilateral)

0.306

0.505

0.655

N

45

43

45

logAPP

correlation coeficient (Spearman’s Rho)

-0.217

0.126

0.283

Sig. (bilateral)

0.158

0.426

0.063

N

44

42

44

  1. Lines 146-150 – this sentence is poorly worded and should be altered to be clearer.

Thank you for the comment and for letting us know that this sentence was unclear. We have rewritten the paragraph (lines 160-166):

“To evaluate whether the magnitude of the differences in transcript expression levels between AD cases and controls was gender-dependent, we included in the multivariate linear regression models the interaction between presence/absence of the disease and gender, resulting in non-statistically significant interactions (circAPP (hsa_circ_0007556), p-value = 0.544 and for APP mRNA, p-value = 0.549). These results demonstrated the absence of sex dependence in the differential expression levels of the transcripts between AD cases and controls.”

  1. The authors use the term ‘lustrum’ on line 86. Despite being a native English speaker, I had to look the word up. I think a different wording here would be easier to follow, and beyond that, it appears that a number of the references referred to are more than 5 years old (so lustrum is not actually an accurate descriptor).

Thanks for the explanation, we have changed it to “In the last few years”

  1. Synaptic is mis-spelled ‘synapsis’ 3 times on lines 295-297.

Thanks for the correction, we have corrected these typos

Reviewer 3 Report

Review for “circRNA from APP gene changes in Alzheimer's disease human brain

 The authors have shown reduced expression of circAPP (hsa_circ_0007556) in AD. They have shown the changes are specific for circAPP whereas the APP mRNA levels were unaltered. Finally the miRNAs that bind to circAPP involving in different pathways were predicted. There are a few points to address:

 1.       Line 28: “join” think its “bind”

2.       Line29: list pathways that are most significant instead of saying “some”

3.       Line 104: mention the “another technique”?

4.       The authors showed circularity of circAPP based on Sanger sequencing of PCR products. However, to assess circularity, RNAse R treatment is normally used. It would be informative to show that the fraction of RNA recovered upon RNase R treatment would be higher for circAPP as compared to linear APP transcript, further confirming the circular nature of circAPP.

5.       It would be informative to provide representative images of Aβ antibody staining and the deposits quantified.

6.       It will be good if the authors could check if any of the predicted miRNAs are upregulated in AD, atleast from published data.

Author Response

Response to the Reviewers’ Comments

We are very grateful of the reviewers’ effort in revising the manuscript and their constructive suggestions.

Review 3

The authors have shown reduced expression of circAPP (hsa_circ_0007556) in AD. They have shown the changes are specific for circAPP whereas the APP mRNA levels were unaltered. Finally the miRNAs that bind to circAPP involving in different pathways were predicted. There are a few points to address:

  1. Line 28: “join” think its “bind”

Thanks for the suggestion. We have changed it.

  1. Line29: list pathways that are most significant instead of saying “some”

Thanks for the suggestion. We have added the 2 most significant pathways (lines 29-30)

  1. Line 104: mention the “another technique”?

 We have changed it to RT-PCR and Sanger sequencing

  1. The authors showed circularity of circAPP based on Sanger sequencing of PCR products. However, to assess circularity, RNAse R treatment is normally used. It would be informative to show that the fraction of RNA recovered upon RNase R treatment would be higher for circAPP as compared to linear APP transcript, further confirming the circular nature of circAPP.

Yes, from a first-time circRNA discovery point of view, the reviewer is absolutely right. Indeed, Mo et al.2020 (1) confirmed the circularity of circAPP by Sanger sequencing after treatment with RNase R the samples. However, once its circular nature was demonstrated by others, our objective was only to try to identify it in our study samples.

  1. It would be informative to provide representative images of Aβ antibody staining and the deposits quantified.

 We have added a figure (Figure S5) in 4.5. Quantitative assessment of Ab deposits in brain tissues

  1. It will be good if the authors could check if any of the predicted miRNAs are upregulated in AD, atleast from published data.

We have added a column in Table 2 with published information on the expression of miRNAs relative to the control and in which tissue/fluid they were studied.

  1. Mo D, Li X, Raabe CA, Rozhdestvensky TS, Skryabin BV, Brosius J. Circular RNA Encoded Amyloid Beta peptides-A Novel Putative Player in Alzheimer's Disease. Cells. 2020;9(10).

Round 2

Reviewer 1 Report

I think revised manuscript would be acceptable in its present form.

Reviewer 2 Report

The authors have addressed most of my concerns and have improved their manuscript. I now support publication.

Reviewer 3 Report

The authors have answered all my queries sufficiently, hence I suggest the study to be accepted for publication.